# Phototactic Changes in *Phthorimaea absoluta* Long-Wavelength Opsin Gene Mutants (*LW2^−/−^*) and Short-Wavelength Opsin Gene Mutant (*BL^−/−^*) Strains

**DOI:** 10.3390/insects15060433

**Published:** 2024-06-07

**Authors:** Yanhong Tang, Xiaodi Wang, Jianyang Guo, Nianwan Yang, Dongfang Ma, Fanghao Wan, Chi Zhang, Zhichuang Lü, Jianying Guo, Wanxue Liu

**Affiliations:** 1State Key Laboratory for Biology of Plant Diseases and Insect Pests, Institute of Plant Protection, Chinese Academy of Agricultural Sciences, Beijing 100193, China; tyanhong2023@163.com (Y.T.); 82101221224@caas.cn (X.W.); guojianyang@caas.cn (J.G.); yangnianwan@caas.cn (N.Y.); wanfanghao@caas.cn (F.W.); liuwanxue@caas.cn (W.L.); 2Engineering Research Center of Ecology and Agricultural Use of Wetland, Ministry of Education, Hubei Collaborative Innovation Center for Grain Industry, College of Agriculture, Yangtze University, Jingzhou 434025, China; madf@yangtzeu.edu.cn; 3Institute of Western Agriculture, The Chinese Academy of Agricultural Sciences, Changji 831100, China; 4Rural Energy and Environment Agency, Ministry of Agriculture and Rural Affairs, Beijing 100125, China; zhangchimy19@163.com

**Keywords:** phototaxis, opsin, *Phthorimaea absoluta*, *LW2*^(−/−)^ and *BL*^(−/−)^ mutant strains

## Abstract

**Simple Summary:**

The tomato leaf miner *Phthorimaea absoluta* is a harmful invasive pest first reported in China in 2017. As an alternative to chemical methods for pest control, recent research has focused on green prevention, specifically light-induced control. However, current light-trapping technology is non-specific, attracting both pests and non-target organisms. Our study aimed to understand the phototactic behavior of *P. absoluta* and to develop a targeted light-trapping technology. Using in situ hybridization, we found widespread opsin expression throughout the insect body. We also investigated tropism using a wavelength-lamp experiment, revealing that 365 ± 5 nm light accurately traps *P. absoluta* without affecting natural enemies. Additionally, we found that long-wavelength opsin gene mutants (*LW2^−/−^*) and short-wavelength opsin gene mutants (*BL^−/−^*) showed significant differences in phototactic behavior. The *LW2^(−/−)^* strain was attracted to light at 390 ± 5 nm, whereas the *BL^(−/−)^* strain was unresponsive. Our findings contribute to the development of precise light-trapping technology for controlling the tomato leaf miner, contributing to our understanding of pest population dynamics and the protection of crops from natural enemies.

**Abstract:**

*Phthorimaea absoluta* (Meyrick) is an invasive pest that has caused damage to tomatoes and other crops in China since 2017. Pest control is mainly based on chemical methods that pose significant threats to food safety and environmental and ecological security. Light-induced control, a green prevention and control technology, has gained attention recently. However, current light-trapping technology is non-specific, attracting targeted pests alongside natural enemies and non-target organisms. In this study, we characterized the phototactic behavior of tomato leaf miners for the development a specific light-trapping technology for pest control. In situ hybridization revealed opsin expression throughout the body. Furthermore, we investigated the tropism of pests (wild *T. absoluta*, *Toxoptera graminum,* and *Bemisia tabaci*) and natural enemies *(Nesidiocoris tenuis* and *Trichogramma pintoi*) using a wavelength-lamp tropism experiment. We found that 365 ± 5 nm light could accurately trap wild *P. absoluta* without trapping natural enemies and other insects. Finally, we analyzed the phototactic behavior of the mutant strains *LW*2^(−/−)^ and *BL*^(−/−)^. *LW2* and *BL* mutants showed significant differences in phototactic behavior. The *LW2*^(−/−)^ strain was attracted to light at 390 ± 5 nm and the *BL^(−/−)^* strain was unresponsive to any light. Our findings will help to develop specific light-trapping technology for controlling tomato leaf miners, providing a basis for understanding pest population dynamics and protecting crops against natural enemies.

## 1. Introduction

*Phthorimaea absoluta* (Meyrick), belonging to the order Lepidoptera, is an invasive agricultural pest introduced into China in recent years [1,2]. Females can lay up to 350 eggs. *P. absoluta* can affect 41 species of plants belonging to 9 families, including vegetables, fruits, food crops, and weeds. The species shows extremely strong adaptability to environmental changes and disasters and can colonize protected areas in cold regions/cold seasons and cause harm in open fields in warm regions/warm seasons [3]. The tomato leaf miner has caused damage in more than 110 countries and regions [4], including Xinjiang, Yunnan, Guizhou, Sichuan, Chongqing, and Beijing, and has substantially impacted tomato, potato, and other industries in China [1,2]. Therefore, there is an urgent need to control the pest. The uncontrolled use of chemicals to control pests has resulted in resistance to several chemicals in tomato leaf miners. Moreover, these chemicals pollute the environment and have a prominent impact on natural enemies (such as *Nesidiocoris tenuis* and *Trichogramma pintoi*) and human health. Developing green control technologies has therefore gained attention.

Trapping, including light trapping and color-plate trapping, is an effective pollution-free method for controlling pests based on insect behavioral responses to light and color [5]. Light trapping controls pests through their phototaxis caused by the light-induced stimulation of visual organs [6]. Zhang et al. [7] used blue–violet light of different wavelengths to trap tomato leaf miners and showed that 380 nm ultraviolet light had the strongest trapping effect. Ardeh et al. [8] found that found that tomato leaf miner is attracted by black light. Zhang et al. [9] used different stick insect colors with sex lures to trap leaf moths and found that the blue trap plate was the most effective. Tan et al. [10] showed that in protected vegetable production, a blue pheromone trap (flat type) placed at a 40–60 cm vent was the most effective. Although light trapping is used to control tomato leaf miners, it will also trap and kill other pests, which will interfere with quantitative statistics and is not conducive to accurately predicting its occurrence dynamics.

The phototactic behavior of insects refers to the stimulation of their visual system by light within a specific spectral range; the photosensitive cells of compound eyes receive the light stimulus and transmit it to the central nervous system, thereby simulating a neural regulatory behavior and phototactic behavior in insects. In the visual system of insects, opsins are vital in visual sensitivity and behavior regulation. Opsins is a member of the G-protein-coupled receptor superfamily and has seven transmembrane helices that participate in signal transduction. In insects, these include long-wavelength-sensitive opsin (*LW opsin*), blue light-sensitive opsin (*BL opsin*), and ultraviolet-sensitive opsin (*UV opsin*) [11,12]. Chen et al. [13] knocked out *LW2 opsin* and established a mutant strain of *Plutella xylostella* with altered phototactic behavior. Chen et al. [4] knocked out *LW2* opsin and observed the loss of *LW2 opsin* sensitivity to a long wavelength. Liu et al. [14] knocked out the *HaBL* and *HaLW* genes using the CRISPR/Cas9 system, leading to a significant decrease in the phototaxis and the climbing of *Helicoverpa armigera*. Various *opsin* mutant strains developed based on genome editing technology show altered visual behavior, such as phototaxis [4,14]. In order to achieve precise trapping, we used the homozygous lines of the constructed opsins combined with light-trapping technology, which will provide a basis for comprehensive pest control. In this study, we aimed to develop an accurate green control technology for target gene pests. The insects used in this experiment included pests (wild *T. absoluta*, *LW2*^(−/−)^
*T. absoluta*, *BL*^(−/−)^ *T. absoluta*, *Toxoptera graminum,* and *Bemisia tabaci*) and natural enemies (*N. tenuis* and *T. pintoi*). Analyses of gene localization, expression, wavelength screening, and phototactic behavior were carried out based on opsin genes and strains in our previous study [15]. Opsins were expressed in the whole body of tomato leaf miners and are vital in phototaxis. Light at 365 ± 5 nm accurately trapped the wild tomato leaf miner. Finally, the light of a specific wavelength could trap a single target gene pest but not natural enemies.

## 2. Materials and Methods

### 2.1. Insect Materials and Rearing

The wild-type tomato leaf miner population used in this study was collected in Yuxi, Yunnan Province, China, in August 2018 and was raised on healthy tomato plants cultivated under greenhouse conditions. *LW2*^(−/−)^ and *BL*^(−/−)^ [15] were obtained through genome editing, and the mutant strains were raised separately under the same conditions. The feeding conditions were as follows: 25 ± 2 °C; 50–60% relative humidity; 14 h of light and 10 h of darkness. Under the same conditions, the host plants were grown alone in pots with a diameter of 9 cm.

*T. graminum* was cultivated from the indoor population raised by the Beijing Plant Protection Institute in an artificial incubator at 20 °C, humidity of 50–70%, and 17 h of light and 7 h of darkness.

*B. tabaci* was obtained from the indoor population raised by the Beijing Plant Protection Institute and were kept in artificial incubators at 26 ± 1 °C, relative humidity of 50–60%, and 14 h of light and 10 h of darkness.

*N. tenuis* was purchased from Beijing Keyun Biotechnology Co., Ltd. (Beijing, China) and raised in an artificial incubator at 25 ± 1 °C, relative humidity of 55–65%, and 14 h of light and 10 h of darkness.

*T. pintoi* was obtained from the indoor population raised by the Plant Protection Institute of Beijing Academy of Agricultural Sciences in an artificial incubator at 26 °C, a relative humidity of 65 ± 5%, and 14 h of light and 10 h of darkness.

### 2.2. RNA In Situ Hybridization

According to protocols described by Ji et al. [16] and Yang et al. [17], the in situ hybridization of adult tomato leaf miners was carried out using a digoxin-labeled ribose probe. The probe primers for in situ hybridization were amplified through polymerase chain reaction (PCR) as follows: Blue opsin-F-CGGCTTCACTATCGG; Blue opsin-R-AAGTAGTCGGTTCCACA. The PCR product was cloned into the pGEM-T Easy vector (Promega, Madison, WI, USA). Antisense and sense RNA probes were synthesized. On the first day, the moths were fixed in Canoy stationary solution (ethanol: chloroform: glacial acetic acid, 6:3:1) at 4 °C overnight. After fixation, they were rinsed in 50% ethanol for 5 min three times and then in 1× phosphate-buffered saline (PBS) at 37 °C for 30 min. Next, they were bleached with 6% hydrogen peroxide in ethanol for 2 h at room temperature (26 °C), washed in 1× PBS, and permeabilized with protease K at 56 °C for 30 min. After washing with 1× PBS, the adults were re-fixed with 4% formaldehyde in 1× PBS at 4 °C and rinsed in 1× PBS three times for 5 min each time. Subsequently, overnight hybridization was performed at 55 °C. Incubation was performed using an anti-DIG antibody (1:5000 dilution; Roche, Basel, Switzerland), followed by nitrotetrazole blue chloride/5-bromo-4-chloro-3-indolyl phosphate in the dark (Roche). Images were captured using a VHX-2000 digital microscope (Keyence, Osaka, Japan).

### 2.3. Identification of Tropism under Different Wavelengths of Light

A polygon behavior device composed of a behavior reaction, dark, static zone, and lighting zones is used to analyze the wavelength-trapping behavior. The experimental setup was composed of acrylic plates (except for the transparent acrylic plates above the polygon, which were opaque milky white). The device was equipped with 16 wavelengths (325 ± 5 nm, 365 ± 5 nm, 375 ± 5 nm, 385 ± 5 nm, 390 ± 5 nm, 395 ± 5 nm, 400 ± 5 nm, 405 ± 5 nm, 415 ± 5 nm, 420 ± 5 nm, 430 ± 5 nm, 450 ± 5 nm, 460 ± 5 nm, 490 ± 5 nm, 520 ± 5 nm, and 590 ± 5 nm) with the same light intensity. The light source manufacturer comes from Shenzhen Fangpu Optoelectronics Co., Ltd. (Shenzhen, China), which provides light sources for the research and development of many university laboratories and enterprises. The lamp source is composed of integrated LED lamp beads of different powers. The chromatographic peak test method is FWHM; the lens power can be 60 degrees, 90 degrees, 120 degrees; and the voltage is 220. A black opaque flannel with a length of 1.5 m and a width of 2 m was used to prevent external light from entering. The experiment was conducted in the dark.

#### 2.3.1. Trapping of Wild *P. absolutaand* Insects in the Same Niche

Before the light-trapping experiment, wild *P. absoluta*, *T. graminum*, *B. tabaci*, *N. tenuis*, and *T. pintoi* were placed in a dark room for 1 h simultaneously, and the drawer board was opened. After they entered the behavioral response area, 16 LED lights with different wavelengths were turned on at the same time, and the partition of the cubicle was opened so that the test insects could enter the cubicle according to their own preferences. The black velvet was then removed, and the number of insects in each compartment was recorded. Each experiment was divided into four groups—wild-type *P. absoluta* adults with *T. graminum*, *B. tabaci*, *N. tenuis*, and *T. pintoi*—with 30 insects per group. All experiments were repeated four times, and the behavioral response device was cleaned after each experiment [4,18].

Phototactic response number = (number of insects in the light chamber/total number of insects used in the experiment).

#### 2.3.2. Trapping of the *LW2^(−/−)^* Strain and Insects in the Same Niche

The wavelength-dependent trapping behaviors of the *LW2*^(−/−)^ strain, *T. graminum*, *B. tabaci*, *N. tenuis*, and *T. pintoi* were evaluated following the methodology described in Section 2.3.

#### 2.3.3. Trapping of the *BL^(−/−)^* Strain and Insects in the Same Niche

The wavelength-dependent trapping behavior of the *BL*^(−/−)^ strain, *T. graminum*, *B. tabaci*, *N. tenuis*, and *T. pintoi* were outlined following the methodology described in Section 2.1. One-way analysis of variance (ANOVA) was used to analyze the number of insects at different wavelengths. Data are presented as the mean ± standard error (mean ± SEM). Differences were considered statistically significant at *p* < 0.05.

#### 2.3.4. Statistical Analysis

Statistical analyses were performed using DPS software (Ruifeng, Hangzhou, China). Differences between groups were analyzed using one-way analysis of variance (ANOVA). Data are presented as the mean ± standard error (mean ± SEM). Differences were considered statistically significant at *p* < 0.05.

## 3. Results

### 3.1. Opsin Expression Profiles in Phthorimaea absoluta

Opsin gene expression in adult *P. absoluta* was studied using a digoxin-labeled RNA ribose probe combined with cell-specific mRNA. The analysis of the antisense probe labeled with digoxin (Figure 1A) showed that opsin was expressed in the whole body of tomato leaf miner, including the head, eyes, and wings. However, there was no expression signal in the negative control (digoxin-labeled sense probe) (Figure 1B). These results show that opsin is vital for the activities of nocturnal leaf moths.

### 3.2. Phototaxis of P. absoluta and Other Insects in the Same Niche under Light of Different Wavelengths

#### 3.2.1. Trapping Effects of Different Wavelengths on Wild Tomato Leaf Miner and Insects in the Same Niche

Wavelength tropism experiments showed that light can be used to trap the wild tomato leaf miner at a wavelength of 365 ± 5 nm, without any effect on natural enemies, such as *N. tenuis* and *T. pintoi* (Table 1), and pests, such as *T. graminum* and *B. tabaci*.

#### 3.2.2. Trapping Effects of Different Wavelengths on *LW2^(−/−)^* and Insects in the Same Niche

Wavelength tropism experiments on *the LW2*^(−/−)^ strain, pests (*T. graminum* and *B. tabaci*), and natural enemies (*N. tenuis* and *T. pintoi*) showed that light at 390 ± 5 nm can significantly trap the strain *LW2*^(−/−)^ but not other insects (Table 2), indicating that *LW2* is vital in the phototaxis of insects. *T. pintoi* was strongly attracted to light at 415 ± 5 nm, whereas *N. tenuis* was photophobic. *T. graminum* was strongly attracted to light at 590 ± 5 nm, and *B. tabaci* was attracted to light at 520 ± 5 nm.

#### 3.2.3. Trapping Effects of Different Wavelengths on *BL^(−/−)^* and Insects in the Same Niche

We performed wavelength tropism experiments on the *BL*^(−/−)^ strain, pests (*T. graminum* and *B. tabaci*), and natural enemies (*N. tenuis* and *T. pintoi*) occurring simultaneously. *BL* was essential in the phototaxis of tomato leaf miners, as the *BL*^(−/−)^ strain was unresponsive to light at any wavelength (Table 3). *T. pintoi* and *N. tenuis* were attracted to light at 415 ± 5 nm and 0 nm, respectively; *T. graminum* and *B. tabaci* were attracted to light at 590 ± 5 nm and 520 ± 5 nm, respectively.

#### 3.2.4. Phototactic Changes in Wild *T. absoluta*, *LW2^(−/−)^ T. absoluta*, and *BL^(−/−)^ P. absoluta* Mutant Strains

Table 4 demonstrates the phototaxis of the wild tomato leaf miner, *LW2^(−/−)^* strain, and *BL*^(−/−)^ strain. The wavelength preferences of the *LW2*^(−/−)^ strain and *BL*^(−/−)^ strain for wavelength changed, and opsin affected the phototaxis of insects, as wild tomato leaf miner was attracted to light at 365 ± 5 nm, the *LW2*^(−/−)^ strain was attracted to light at 390 ± 5 nm, and the *BL*^(−/−)^ strain was unresponsive to light at any wavelength.

## 4. Discussion

Opsins are important factors in the phototactic behavior of insects. In the compound eyes of insects, opsin is involved in light transmission [19] and perception [20,21]. Opsins are highly expressed in the compound eyes of insects, such as *H. armigera*, *Manduca sexta*, *Drosophila melanogaster,* and *N. tenuis* [21,22,23,24]. Studies have shown that opsins are expressed in other tissues [25]. For example, in green-blind stinkbugs, opsins are highly expressed in the head, legs, wings, and mouthparts [26]; in *D. melanogaster*, opsins are highly expressed in the antennae and head [27]. In bees and *H. armigera*, opsins are expressed in the brain [28]. In our study, we found that opsin is expressed throughout the body parts of moth, as seen in other studies. As nocturnal insects, tomato leaf miners, in order to adapt to the low-light environment, evolved a sensory mechanism and may be able to use the whole-body expression of visual genes to respond to target objects quickly at night to avoid and prey, which may be a key characteristic of nocturnal insects.

In recent years, many entomologists have studied the sensitivity spectra and phototactic behaviors of Coleoptera, Lepidoptera, Hymenoptera, Hemiptera, and Neuroptera through indoor and outdoor behavioral assays and visual electrophysiology. They found that most insects prefer ultraviolet, blue, and green light [29]. Moreover, many natural enemies have similar areas of spectral sensitivity. Thus, natural enemies and other non-target insects are targeted when using insect trap lamps. This poses a threat to biodiversity and the natural ecological balance. Zhang et al. [7] showed that tomato leaf miners tend to be exposed to blue and violet light; amongst these, 380 nm and 405 nm ultraviolet light traps can effectively attract the species alongside natural enemies. Shiberu et al. [30] showed that tomato leaf miners have different tendencies toward different colors, among which white and blue swatches were most suitable for monitoring leaf moths in greenhouses. However, white and blue plates can also attract other insects, which is not conducive for their accurate prevention and control. Yang et al. [31] showed that natural enemies and neutral insects are sometimes killed when using light-trapping technology to control pests, mainly because many pests in the field are mixed with natural enemies. However, the phototaxis spectra differ among pests. Therefore, a 365 ± 5 nm ultraviolet LED lamp can accurately trap the wild tomato leaf miner but not the natural enemy insects. This provides a theoretical basis for the accurate prevention and control of tomato leaf miner.

Silencing and knocking out *opsin* genes using RNAi or CRISPR/Cas9 technology can significantly affect the phototactic behavior of insects, such as *Spodoptera exigua*, *H. armigera*, and *Plutella xylostella* [4,18,32]. However, no opsin-mutant strains have been used for green prevention or control. Based on the opsin strains generated in the laboratory, we tested the trapping of *LW2*^(−/−)^ and *BL*^(−/−)^ strains along with insects in the same niche. The results showed that the *LW2*^(−/−)^ strain was attracted by 390 ± 5 nm light. The wavelength range of LW2 is 500–600 nm [11]. In our study, we found that *LW2*^(−/−)^ was strongly attracted by light at 390 nm, indicating that the *LW2* gene plays an important role in insect phototaxis. Previous studies have shown that changes in the opsin spectrum are due to changes in amino acids that affect the spectral regulation of opsin [33,34]. In our study, in the *LW2*^(−/−)^ strain, the conversion of a threonine (Thr) to alanine (Ala) affects its spectral adjustment. When we tested the trapping situation of the *BL^(−/−)^* strain and insects in the same niche, we found that *BL^(−/−)^* strain was unresponsive to light at any wavelength. In addition, the knockout of *Ci-opsin1* renders insects unresponsive to light [35]. Xiong et al. [36] showed that disrupting the maturation, transport, activation, and degradation of Drosophila *Rh1* leads to retinal degeneration and eventually the complete loss of vision. In our experiment, *BL^(−/−)^* strain lost its preference for light, as the visual perception ability was affected. However, the specific mechanism underlying this effect is unclear, and further research is needed.

We did not evaluate all insects within the same niche; in the future, more species should be evaluated. In summary, opsin is a key factor affecting the phototactic behavior of tomato leaf miners. Our results provide a theoretical basis for accurate green prevention, control, and prediction.

## Figures and Tables

**Figure 1 insects-15-00433-f001:**
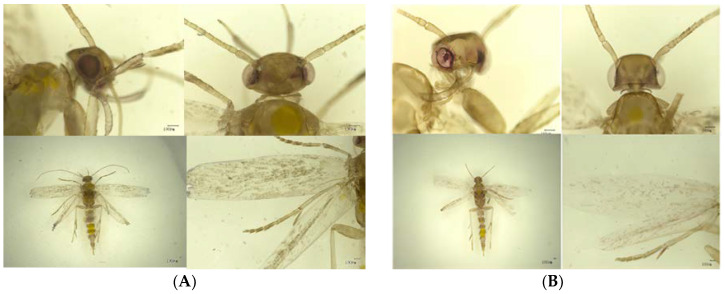
Expression analysis of *Phthorimaea absoluta BL* opsin via in situ hybridization (**A**) Lateral view of an adult hybridized using the opsin antisense riboprobe, showing positive signals in the posterior region of the body. (**B**) Adult tomato leaf miners were hybridized using the opsin sense riboprobe as controls.

**Table 1 insects-15-00433-t001:** Phototaxis of wild tomato leaf miner and insects in the same ecological niche. The data in the table refer to the average of four replicates of each group of experiments.

Wavelength	*T. absoluta*	*T. pintoi*	*N. tenuis*	*T. graminum*	*B. tabaci*
0 nm	3.66 ± 0.47(a)	0.00	26.17 ± 0.47(a)	2.00 ± 0.66(b)	0.00
325 ± 5 nm	0.00	0.00	0.00	0.00	2.66 ± 0.47(c)
365 ± 5 nm	17.66 ± 2.05(a)	0.00	0.00	0.00	0.00
370 ± 5 nm	0.00	0.00	0.00	0.00	0.00
385 ± 5 nm	1.33 ± 0.47(b)	0.00	0.00	0.00	0.00
390 ± 5 nm	0.00	0.00	0.00	0.00	0.00
395 ± 5 nm	0.00	0.00	0.00	0.00	0.00
400 ± 5 nm	0.00	0.00	1.33 ± 0.47(b)	0.00	0.00
405 ± 5 nm	5.00 ± 0.81(b)	0.00	0.00	2.00 ± 0.81(b)	0.00
415 ± 5 nm	0.00	14.00 ± 0.94(a)	1.53 ± 0.50(b)	0.00	0.00
420 ± 5 nm	0.00	8.33 ± 1.25(b)	0.00	0.00	0.00
430 ± 5 nm	0.00	1.30 ± 0.50(c)	0.00	2.00 ± 0.81(b)	0.00
450 ± 5 nm	0.00	0.00	1.00 ± 0.12(b)	5.33 ± 0.47(b)	7.33 ± 1.24(b)
460 ± 5 nm	1.00 ± 0.12(b)	0.00	0.00	0.00	0.00
490 ± 5 nm	2.33 ± 1.24(b)	0.00	0.00	0.00	6.66 ± 0.47(b)
520 ± 5 nm	0.00	4.10 ± 0.81(c)	0.00	5.00 ± 0.81(b)	13.33 ± 1.24(a)
590 ± 5 nm	0.00	1.26 ± 0.47(c)	0.00	13.66 ± 0.94(b)	0.00

Note: The lowercase letters in each column represents significant differences (*p* < 0.05). The data in the table represent the average number of insects at different wavelengths. Data are presented as the mean ± SE. Data were compared by analysis of variance (ANOVA) followed by Tukey’s post hoc test. *P. absoluta* indicates a wild population. (a)–(c) indicate significant differences in the same group.

**Table 2 insects-15-00433-t002:** Phototaxis of the *LW2*^(−/−)^ mutant strain and insects in the same ecological niche. The data in the table refer to the average of four replicates of each group of experiments.

Wavelength	*LW2* * ^(−/−)^ *	*T. pintoi*	*N. tenuis*	*T. graminum*	*B. tabaci*
0 nm	1.66 ± 0.47(b)	0.00	25.17 ± 0.47(a)	2.00 ± 0.66(b)	0.00
325 ± 5 nm	0.00	0.00	0.00	0.00	2.66 ± 0.47(c)
365 ± 5 nm	2.00 ± 0.81(b)	0.00	0.00	0.00	0.00
370 ± 5 nm	0.00	0.00	0.00	0.00	0.00
385 ± 5 nm	1.00 ± 0.43(b)	0.00	0.00	0.00	0.00
390 ± 5 nm	15.00 ± 1.63(a)	0.00	0.00	0.00	0.00
395 ± 5 nm	0.00	0.00	0.00	0.00	0.00
400 ± 5 nm	1.33 ± 0.81(b)	0.00	1.33 ± 0.47(b)	0.00	0.00
405 ± 5 nm	0.00	0.00	0.00	2.00 ± 0.64(b)	0.00
415 ± 5 nm	0.00	15.00 ± 1.04(a)	2.5 ± 0.62(b)	0.00	0.00
420 ± 5 nm	3.33 ± 0.94(b)	7.33 ± 1.25(b)	0.00	0.00	0.00
430 ± 5 nm	0.00	1.30 ± 0.50(c)	0.00	3.00 ± 0.67(b)	0.00
450 ± 5 nm	0.00	0.00	1.00 ± 0.12(b)	4.33 ± 0.36(b)	7.33 ± 1.24(b)
460 ± 5 nm	1.66 ± 0.94(b)	0.00	0.00	0.00	0.00
490 ± 5 nm	2.00 ± 0.81(b)	0.00	0.00	0.00	6.86 ± 0.47(b)
520 ± 5 nm	0.00	3.10 ± 0.81(c)	0.00	4.00 ± 0.81(b)	13.13 ± 1.34(a)
590 ± 5 nm	0.00	2.26 ± 0.67(c)	0.00	14.76 ± 0.94(a)	0.00

Note: The lowercase letters in each column represent significant differences (*p* < 0.05). The data in the table represent the average number of insects at different wavelengths. Data are presented as the mean ± SE. Data were compared by analysis of variance (ANOVA) followed by Tukey’s post hoc test. *LW2*^(−/−)^ indicates a mutant line. (a)–(c) indicate significant differences in the same group.

**Table 3 insects-15-00433-t003:** Phototaxis of the *BL*^(−/−)^ mutant strain and insects in the same ecological niche. The data in the table refer to the average of four replicates of each group of experiments.

Wavelength	*BL* * ^(−/−)^ *	*T. pintoi*	*N. tenuis*	*T. graminum*	*B. tabaci*
0 nm	27 ± 1.47(a)	0.00	25.67 ± 0.47(a)	2.00 ± 0.66(b)	0.00
325 ± 5 nm	0.00	0.00	0.00	0.00	3.66 ± 0.47(c)
365 ± 5 nm	1.66 ± 0.47(b)	0.00	0.00	0.00	0.00
370 ± 5 nm	0.00	0.00	0.00	0.00	0.00
385 ± 5 nm	0.00	0.00	0.00	0.00	0.00
390 ± 5 nm	1.34 ± 0.47(b)	0.00	0.00	0.00	0.00
395 ± 5 nm	0.00	0.00	0.00	0.00	0.00
400 ± 5 nm	0.00	0.00	1.33 ± 0.47(b)	0.00	0.00
405 ± 5 nm	0.00	0.00	0.00	2.00 ± 0.81(b)	0.00
415 ± 5 nm	0.00	14.55 ± 0.94(a)	2.00 ± 0.50(b)	0.00	0.00
420 ± 5 nm	0.00	8.33 ± 1.25(a)	0.00	0.00	0.00
430 ± 5 nm	0.00	1.30 ± 0.50(b)	0.00	2.00 ± 0.81(b)	0.00
450 ± 5 nm	0.00	0.00	1.00 ± 0.12(b)	5.33 ± 0.47(b)	7.33 ± 1.24(b)
460 ± 5 nm	0.00	0.00	0.00	0.00	0.00
490 ± 5 nm	0.00	0.00	0.00	0.00	4.66 ± 0.47(c)
520 ± 5 nm	0.00	3.55 ± 0.73(b)	0.00	3.00 ± 0.54(b)	14.33 ± 1.24(a)
590 ± 5 nm	0.00	1.26 ± 0.47(b)	0.00	15.66 ± 1.23(a)	0.00

Note: The lowercase letters in each column represent significant differences (*p* < 0.05). The data in the table represent the average number of insects at different wavelengths. Data are presented as the mean ± SE. Data were compared by analysis of variance (ANOVA) followed by Tukey’s post hoc test. *BL*^(−/−)^ indicates a mutant line. (a)–(c) indicate significant differences in the same group.

**Table 4 insects-15-00433-t004:** Phototaxis of different tomato leaf miner strains. The data in the table refer to the average of four replicates of each group of experiments.

Wavelength	*P. absoluta*	*LW2* * ^(−/−)^ *	*BL* * ^(−/−)^ *
0 nm	3.66 ± 0.47(b)	1.66 ± 0.47(b)	27 ± 1.47(a)
325 ± 5 nm	0.00	0.00	0.00
365 ± 5 nm	17.66 ± 2.05(a)	2.00 ± 0.81(b)	1.66 ± 0.47(b)
370 ± 5 nm	0.00	0.00	0.00
385 ± 5 nm	1.33 ± 0.47(b)	1.00 ± 0.43(b)	0.00
390 ± 5 nm	0.00	15.00 ± 1.63(a)	1.34 ± 0.47(b)
395 ± 5 nm	0.00	0.00	0.00
400 ± 5 nm	0.00	1.33 ± 0.81(b)	0.00
405 ± 5 nm	5.00 ± 0.81(b)	0.00	0.00
415 ± 5 nm	0.00	0.00	0.00
420 ± 5 nm	0.00	3.33 ± 0.94(b)	0.00
430 ± 5 nm	0.00	0.00	0.00
450 ± 5 nm	0.00	0.00	0.00
460 ± 5 nm	1.00 ± 0.12(b)	1.66 ± 0.94(b)	0.00
490 ± 5 nm	2.33 ± 1.24(b)	2.00 ± 0.81(b)	0.00
520 ± 5 nm	0.00	0.00	0.00
590 ± 5 nm	0.00	0.00	0.00

Note: The lowercase letters in each column represents significant differences (*p* < 0.05). The data in the table represent the average number of insects at different wavelengths. Data are presented as the mean ± SE. Data were compared by analysis of variance (ANOVA) followed by Tukey’s post hoc test. *P. absoluta* is the wild-type species. *LW2*^(−/−)^ indicates a mutant line. *BL*^(−/−)^ indicates a mutant line. (a)–(b) indicate significant differences in the same group.

## Data Availability

Data are contained within the article.

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
