# Peer review of "Phototactic Changes in Phthorimaea absoluta Long-Wavelength Opsin Gene Mutants (LW2−/−) and Short-Wavelength Opsin Gene Mutant (BL−/−) Strains"

_insects, 2024, doi:10.3390/insects15060433_

Round 1

Reviewer 1 Report

Comments and Suggestions for Authors

In the present manuscript, the authors explored the precise light-trapping behavior of Tuta absoluta and its opsin-mutant strains, as well as the pest species and natural enemies living in the same habitat as T. absoluta. The findings provide the foundation to develop an effective light trapping technology for controlling tomato leaf miners. Comments and suggestions are given below to be addressed.

Major comments

1.        The simple summary and abstract are the same. I suggest authors rewrite simple summary which would not be repeated in abstract.

2.        Please provide the reference or reason for the primers used for the probe to detect the expression of opsin. In addition, which opsin to be detected by the probe is not clear since there are UV-opsin, Blue-opsin, LW-opsin in insects.

3.        In lines 179-181, the authors claimed that opsin was expressed in the whole body of tomato leaf miner, such as head, eyes, wings and so on. It is easy to see any differences in opsin expressions in Fig. 1 A&B. Please provide a clear picture to determine the expressions of opsin in tomato leaf miner.

4.        Why were only these (T. absoluta, T. graminum, B. tabaci, N. tenuis, T. pintoi ) species selected as a non-target? There must be a lot of species available in the same habitat with wild tomato leaf miner.

5.        In Lines 91-96, the objective and hypothesis are not clear, and please rewrite the objective and hypothesis. Also, please focus on your objective in results and discussion section. I have noticed the conclusion made by authors is not following the objective of study.

Minor comments

6.        Provide details about BL (-/-) and LW2(-/-) when it first appears in the text.

7.        In line 89, gene editing should be genome editing.

8.        In line 97, cite the reference.

9.        In lines 105-108, what do you mean “base substitution”? Better to remove these sentences and just cite reference which indicates the knockout of these two genes.

10.    In lines 171-174, provide details of “ANOVA and multiple comparison test”.

11.    Better to use the scientific names of species in all text rather than to mix with common names.

12.    In line 195, which ANOVA test is used, one-way or two-way?

13.    “3.1. RNA in situ hybridization” need to be changed to “3.1. Opsin expression profiles in Tuta absoluta”

14.    “3.2. Trapping of different wavelengths on the same niche of T. absoluta” need to be changed to “3.2. Phototaxis of T. absoluta and other insects in the same niche to the lights of different wavelengths”

15.    Please check all tables and highlight significant differences with lowercase letters.

16.    Caption of all tables should be revised like “The lowercase letters in each column represents”

Comments on the Quality of English Language

The manuscript must be revised by native English speaker because there is a lot of grammatical errors.

Author Response

Responses to reviewers 1

May 6, 2024

We have read and studied the reviewers’ comments and suggestions very carefully. We have revised our manuscript based on the reviewers' comments. The main corrections in the paper and the responses to the reviewer’s comments are as follows.

Responses to Editor

Comments to the Author:

The manuscript presents results on phototaxis of tomato leaf-miner moths (including some opsin mutants) as a way to implement traps against this tomato pest. While the topic justifies peer-review, the text requires careful English revisions. For instance in the abstract, the authors write that opsin mutants attract light when in fact it is the opposite, they are attracted by light. This kind of mistake is recurrent in the text and requires careful revision by native English speaker.

Response: Thanks for your comments. We have revised the full text, and let the special article retouching company make the revision. The touch-up certificate is as follows:

Responses to Reviewer 1

Comments to the Author:

In the present manuscript, the authors explored the precise light-trapping behavior of Tuta absoluta and its opsin-mutant strains, as well as the pest species and natural enemies living in the same habitat as T. absoluta. The findings provide the foundation to develop an effective light trapping technology for controlling tomato leaf miners. Comments and suggestions are given below to be addressed.

Major comments

Q1. The simple summary and abstract are the same. I suggest authors rewrite simple summary which would not be repeated in abstract.

Response: Thanks for your comments. We rewrite the simple summary,please check from line 12 to 26.The contents are as follows: The tomato leaf miner Phthorimaea absoluta is a harmful invasive pest first reported in China in 2017. As an alternative to chemical methods for pest control, recent research has focused on green prevention, specifically light-induced control. However, current light-trapping technology is non-specific, attracting both pests and non-target organisms. Our study aimed to understand the phototactic behavior of P.absoluta and to develop a targeted light-trapping technology. Using in situ hybridization, we found widespread opsin expression throughout the insect body. We also investigated tropism using a wavelength-lamp experiment, revealing that 365 ± 5 nm light accurately traps P. absoluta without affecting natural enemies. Additionally, we found that the mutant strains LW2(-/-) and BL (-/-) showed significant differences in phototactic behavior. The LW2(-/-) strain was attracted to light at 390 ± 5 nm, whereas the BL (-/-) strain was unresponsive. Our findings contribute to the development of precise light trapping technology for controlling the tomato leaf miner, contributing to our understanding of pest population dynamics and the protection of crops from natural enemies.

Q2. Please provide the reference or reason for the primers used for the probe to detect the expression of opsin. In addition, which opsin to be detected by the probe is not clear since there are UV-opsin, Blue-opsin, LW-opsin in insects.

Response: Based the following three references, the primers of the probe are used to detect the expression of opsin: firstly, the sequence of the primer is specific; secondly, the length is less than 500, and thirdly, the GC content of the probe is preferably 40-60%. In addition, the Blue-opsin was detected by the probe in the study, and the information had been added to the revised manuscript, and check it in lines 124, 185.

Q3. In lines 179-181, the authors claimed that opsin was expressed in the whole body of tomato leaf miner, such as head, eyes, wings and so on. It is easy to see any differences in opsin expressions in Fig. 1 A&B. Please provide a clear picture to determine the expressions of opsin in tomato leaf miner.

Response: We re-uploaded Fig. 1 A&B. The Figure 1 is as follows:

Figure 1. Expression analysis of Tuta absoluta BL opsin via in situ hybridization (A) Lateral view of an adult hybridized using the opsin antisense riboprobe, showing positive signals in the posterior region of the body. (B) Adult tomato leaf miners were hybridized using the opsin sense riboprobe as controls.

Q4. Why were only these (T. absoluta, T. graminum, B. tabaci, N. tenuis, T. pintoi) species selected as a non-target? There must be a lot of species available in the same habitat with wild tomato leaf miner.

Response: Thanks for your comments. There were many insects that occurred at the same time as the tomato leaf miner, generally, it was found that Toxoptera graminum and Bemisia tabaci were more serious than other insects, and Nesidiocoris tenuis and Trichogramma pintoi, as natural enemies, were often used to control the tomato leaf miner. Therefore, we chose these four insects.

Q5. In Lines 91-96, the objective and hypothesis are not clear, and please rewrite the objective and hypothesis. Also, please focus on your objective in results and discussion section. I have noticed the conclusion made by authors is not following the objective of study.

Response: Thanks for your comments. Our goals are: Although light trapping is used to control tomato leaf miner, it will also trap and kill other pests, which will interfere with the quantitative statistics and is not conducive to accurately predicting its occurrence dynamics, and please check lines from 70 to 73. In order to achieve precise trapping, we used the homozygous lines of the constructed opsins to combined with light trapping technology, and will provide a basis for comprehensive pest control, and please check lines from 88 to 90.

In the discussion part, we discussed the two points we put forward, for example, in recent years, many entomologists have studied the sensitivity spectra and phototactic behaviors of Coleoptera, Lepidoptera, Hymenoptera, Hemiptera, and Neuroptera through indoor and outdoor behavioral assays and visual electrophysiology. They found that most insects prefer ultraviolet, blue, and green light [29]. Moreover, many natural enemies have similar areas of spectral sensitivity. Thus, natural enemies and other non-target insects are targeted when using insect trap lamps. This poses a threat to biodiversity and the natural ecological balance. Zhang et al. [7] showed that tomato leaf miners tend to be exposed to blue and violet light; amongst these, 380 nm and 405 nm ultraviolet light traps can effectively attract the species alongside natural enemies. Shiberu et al. [30] showed that tomato leaf miners have different tendencies toward different colors, among which white and blue swatches were most suitable for monitoring leaf moths in greenhouses. However, white and blue plates can also attract other insects, which is not conducive for their accurate prevention and control. Yang et al. [31] showed that natural enemies and neutral insects are sometimes killed when using light-trapping technology to control pests, mainly because many pests in the field are mixed with natural enemies. However, the phototaxis spectra differ among pests. Therefore, a 365 ± 5 nm ultraviolet LED lamp can accurately trap the wild tomato leaf miner but not the natural enemy insects. This provides a theoretical basis for the accurate prevention and control of tomato leaf miner. Please check from lines 254 to 272.

Silencing and knocking out opsin genes using RNAi or CRISPR/Cas9 technology can significantly affect the phototactic behavior of insects, such as Spodoptera exigua, H. armigera, and Plutella xylostella [4,18,32]. However, no opsin-mutant strains have been used for green prevention or control. Based on the opsin strains generated in the laboratory, we tested the trapping of LW2(-/-) and BL (-/-) strains along with insects in the same niche. The results showed that the LW2(-/-) strain was attracted by 390 ± 5 nm light. The wavelength range of LW2 is 500–600 nm [11]. In our study, we found that LW2(-/-) was strongly attracted by light at 390 nm, indicating that the LW2 gene plays an important role in insect phototaxis. Previous studies have shown that changes in the opsin spectrum are due to changes in amino acids that affect the spectral regulation of opsin [33,34]. In our study, in the LW2(-/-) strain, the conversion of a threonine (Thr) to alanine (Ala) affects its spectral adjustment. When we tested the trapping situation of the BL (-/-) strain and insects in the same niche, we found that BL (-/-) strain was unresponsive to light at any wavelength. In addition, the knockout of Ci-opsin1 renders insects unresponsive to light [35]. Xiong et al. [36] showed that disrupting the maturation, transport, activation, and degradation of Drosophila Rh1 leads to retinal degeneration and eventually the complete loss of vision. In our experiment, BL (-/-) strain lost its preference for light, as the visual perception ability was affected. However, the specific mechanism underlying this effect is unclear, and further research is needed, please check lines from 272 to 290.

Minor comments

Q6. Provide details about BL (-/-) and LW2(-/-) when it first appears in the text.

Response: We changed BL (-/-) and LW2(-/-) to blue-wavelength opsin gene mutants (BL-/-) and Long-wavelength opsin gene mutants (LW2-/-), respectively, please check from lines 22 to 23.

Q7. In line 89, gene editing should be genome editing.

Response: Thanks for your comments. We have changed gene editing to genome editing, please check in line 87, 103.

Q8. In line 97, cite the reference.

Response: Thanks for your comments. We have added relevant literature in this section, please check in line 103. The reference is as follow:

Tang, Y.H.; Bi, S.Y.; Wang, X.D.; Ji, S.X.; Huang, C.; Zhang, G.; Guo, J.; Yang, N.; Ma, D.; Wan, F.; Lü, Z.; Liu, W. Opsin mutants alter host plant selection by color vision in the nocturnal invasive pest Tuta absoluta. International journal of biological macromolecules. 2024, 130636.

Q9. In lines 105-108, what do you mean “base substitution”? Better to remove these sentences and just cite reference which indicates the knockout of these two genes.

Response: Thanks for your comments. “Base substitution” means “basepair substitution”. We deleted this part and quoted relevant literature, please check from lines103 to 104. The reference is as follows:

Tang, Y.H.; Bi, S.Y.; Wang, X.D.; Ji, S.X.; Huang, C.; Zhang, G.; Guo, J.; Yang, N.; Ma, D.; Wan, F.; Lü, Z.; Liu, W. Opsin mutants alter host plant selection by color vision in the nocturnal invasive pest Tuta absoluta. International journal of biological macromolecules. 2024, 130636.

Q10.  In lines 171-174, provide details of “ANOVA and multiple comparison test”.

Response: Thanks for your comments. We added relevant information as follows: One-way analysis of variance (ANOVA) was used to analyze the number of insects at different wavelengths. Data are presented as the mean ± standard error (mean ± SEM). Differences were con-sidered statistically significant at P < 0.05. Please check from lines 168 to 170.

Q11.   Better to use the scientific names of species in all text rather than to mix with common names.

Response: Thanks for your comments. We carefully checked the full text and made some changes, please check lines in 28, 44, 47.

Q12.    In line 195, which ANOVA test is used, one-way or two-way?

Response: Thanks for your comments. We used one-way ANOVA test.

Q13. “3.1. RNA in situ hybridization” need to be changed to “3.1. Opsin expression profiles in Tuta absoluta

Response: Thanks for your comments. We changed “3.1. RNA in situ hybridization” to “3.1. Opsin expression profiles in Tuta absoluta”. Please check in line 177.

Q14. “3.2. Trapping of different wavelengths on the same niche of T. absoluta” need to be changed to “3.2. Phototaxis of T. absoluta and other insects in the same niche to the lights of different wavelengths.

Response: Thanks for your comments. We changed“3.2. Trapping of different wavelengths on the same niche of T. absoluta” need to be changed to “3.2. Phototaxis of T. absoluta and other insects in the same niche to the lights of different wavelengths”. Please check from line189 to 190.

Q15.  Please check all tables and highlight significant differences with lowercase letters.

Response: Thanks for your comments. We use lowercase letters to indicate the significant differences in all tables of the article. Please check line in 197, 210, 223, 236.

Q16.   Caption of all tables should be revised like “The lowercase letters in each column represents”

Response: Thanks for your comments. We made some changes, please check lines in197, 210, 223, 236.

Q17. The manuscript must be revised by native English speaker because there is a lot of grammatical errors.

Response: Thanks for your comments. We have revised the full text, and let the special article retouching company make the revision. The touch-up certificate is attached.

Thank you once again for your profound comments and questions. We hope our corrections meet your requirements. We are very grateful to you for your kind help in processing this manuscript. We look forward to seeing its publication in your journal soon.

Reviewer 2 Report

Comments and Suggestions for Authors

This study explores the phototactic behavior of an important pest and green methods (i.e. not involving any pollutants) to trap them. In fact, the authors are too shy, as in fact they tested more species. The study is both interesting and important, and I enjoyed reading it. However, I feel it could benefit from getting more detailed in some places (see below).

The largest concern I have is about the actual characteristics of the light sources.

L141-142: 16 wavelengths (325 ± 5 nm, 365 ± 5 nm, 375 ± 5 nm, 385 ± 5 nm, 390 ± 5 nm, 395 ± 5 nm, 400 ± 5 nm, 405 ± 5 nm, 415 nm ± 5, 420 ± 5 nm, 430 ± 5 nm, 450 ± 5 nm, 460 ± 5 nm, 490 ± 5 nm, 520 ± 5 nm, and 590 ± 5 nm): this sounds not very convincing.

First, please provide some information about the producer and technical characteristics of the light sources.

Second, what is ± 5 nm? It sounds very improbable that a LED device would give such a narrow peak. Is that FWHM? More plausible but still needs checking.

Third, if these ± 5 nm were stated by the manufacturer, please check the real spectrum with a spectrometer and state the real data. The same applies to the light intensity (you say it was equal, but did you measure it?).

Then, I feel that the work could really benefit from more detailed description of the information given in the main table.

Table 1: I don’t completely understand how these numbers are formed. According to the methods section above, the numbers should be a percent out of 30 (the total number of insects in each experiment). How are the number like 1.00 and 2.00 percent obtained, as well as other numbers which are not some share of 30? Or are these numbers mean numbers of insects (as for me, it would be more logical)? Please state more clearly what the numbers mean and add this statement to the caption for the table.

In addition, I find it really strange that 370 ± 5 nm was so different from 365 ± 5 nm. Even if these light sources are so focused, the insect’s visual opsins are hardly so. Do you have any hypotheses why that might have happened?

Did you repeat the experiments for T. pintoi, N. tenuis, T. graminum and B. tabaci anew for each table? It’s really interesting that the results are extremely similar but yet adjacent wavelengths differ a lot. Again, do you maybe have a hypothesis explaining this?

Finally, I don’t feel a strong connection between the information given in the first figure and the rest of the paper, even though I totally agree that this information is important. But why do the authors draw the conclusion about the importance of the opsin protein(s) from the fact they are expressed in different parts of the body? It makes some sense but it not entirely clear. It’s also interesting to elaborate (maybe in the discussion) if the authors suggest that the opsins expressed extraocularly have a role in photoreception and attraction by light. Is there such evidence for the studied or related species?

Comments on the Quality of English Language

L21 “Long-wavelength” should be lowercase (long-wavelength)

L65 “found that tomato leaf miner attracts black light” => “found that tomato leaf miner is attracted by black light”

L78-79: opsinS (as there may be more than one in each class)

L88 “pest control In this study” : full stop missing

L176 “Digoxin-labeled” should be “digoxin-labeled”

Tables: please make species names italic

L236 and below: “Opsin is highly expressed”: I would say opsins, as these are all different representatives of this protein group.

Author Response

Responses to reviewers 2

May 6, 2024

We have read and studied the reviewers’ comments and suggestions very carefully. We have revised our manuscript based on the reviewers' comments. The main corrections in the paper and the responses to the reviewer’s comments are as follows.

Responses to Reviewer 2

Comments to the Author:

This study explores the phototactic behavior of an important pest and green methods (i.e. not involving any pollutants) to trap them. In fact, the authors are too shy, as in fact they tested more species. The study is both interesting and important, and I enjoyed reading it. However, I feel it could benefit from getting more detailed in some places (see below).

Q1. The largest concern I have is about the actual characteristics of the light sources.

L141-142: 16 wavelengths (325 ± 5 nm, 365 ± 5 nm, 375 ± 5 nm, 385 ± 5 nm, 390 ± 5 nm, 395 ± 5 nm, 400 ± 5 nm, 405 ± 5 nm, 415 nm ± 5, 420 ± 5 nm, 430 ± 5 nm, 450 ± 5 nm, 460 ± 5 nm, 490 ± 5 nm, 520 ± 5 nm, and 590 ± 5 nm): this sounds not very convincing.

First, please provide some information about the producer and technical characteristics of the light sources. 

Response: Thanks for your comments. The light source manufacturer comes from Shenzhen Fangpu Optoelectronics Co., Ltd., which provides light sources for the research and development of many university laboratories and enterprises. The lamp source is composed of integrated LED lamp beads of different powers, the peak wavelength can be 365 ± 5, 370±5, 380±5nm, etc., the lens power can be 60 degrees, 90 degrees, 120 degrees, and the voltage is 220.

Q2. Second, what is ± 5 nm? It sounds very improbable that a LED device would give such a narrow peak. Is that FWHM? More plausible but still needs checking.

Response: The peak wavelength fluctuates within 5 nm. For example, the peak wavelength of 425 is between 420 and 425 nm. After consulting with the company that provides the light source, it should be FWHM. The test peak diagram is as follows:

Q3. Third, if these ± 5 nm were stated by the manufacturer, please check the real spectrum with a spectrometer and state the real data. The same applies to the light intensity (you say it was equal, but did you measure it?).

Response: For example, the company provided a peak diagram of light with a wavelength of 420±5nm, with a peak wavelength of 421.5nm and a voltage of 30w, as shown in the figure:

Q4. Then, I feel that the work could really benefit from more detailed description of the information given in the main table.

Table 1: I don’t completely understand how these numbers are formed. According to the methods section above, the numbers should be a percent out of 30 (the total number of insects in each experiment). How are the number like 1.00 and 2.00 percent obtained, as well as other numbers which are not some share of 30? Or are these numbers mean numbers of insects (as for me, it would be more logical)? Please state more clearly what the numbers mean and add this statement to the caption for the table. 

Response: Thanks for your comments. The data in Table 1 refers to the average of four replicates of each group of experiments. This statement has been added to the title of the table, please check it in line 196, 209, 223, and 235.

Q5. In addition, I find it really strange that 370 ± 5 nm was so different from 365 ± 5 nm. Even if these light sources are so focused, the insect’s visual opsins are hardly so. Do you have any hypotheses why that might have happened?

Response: Thanks for your comments. The insect's visual system is very complex. The light passes through the complex compound eye structure of the moth and is finally perceived by the visual pigment in the rod. The visual pigment can generate biological potential for the spectrum in a certain wavelength range and transmit it to the central nervous system to cause visual response. Visual pigment consists of visual protein and photosensitive chromophore, and most organisms only can synthesize a single chromophore, and has the diversity of visual pigment absorption spectrum. Depending on the selective substitution of key amino acids in opsin, this may lead to the formation of different photoreceptors. The selective substitution of key amino acids and the formation of different photoreceptors may be one of the reasons for the different phototaxis.

Q6. Did you repeat the experiments for T. pintoi, N. tenuis, T. graminum and B. tabaci anew for each table? It’s really interesting that the results are extremely similar but yet adjacent wavelengths differ a lot. Again, do you maybe have a hypothesis explaining this?

Response: Thanks for your comments. We conducted the experiments in Table 1, Table 2 and Table 3 respectively. In Table 1, before the light-trapping experiment, wild P. absoluta, T. graminum, B. tabaci, N. tenuis, and T. pintoi were placed in a dark room for 1 h simultaneously, and the drawer board was opened. After entering the behavioral response area, 16 LED lights (325 ± 5 nm, 365 ± 5 nm, 375 ± 5 nm, 385 ± 5 nm, 390 ± 5 nm, 395 ± 5 nm, 400 ± 5 nm, 405 ± 5 nm, 415 nm ± 5, 420 ± 5 nm, 430 ± 5 nm, 450 ± 5 nm, 460 ± 5 nm, 490 ± 5 nm, 520 ± 5 nm, and 590 ± 5 nm) with different wavelengths were turned on at the same time, and the partition of the cubicle was opened so that the test insects could enter the cubicle according to their own pref-erences. The black velvet was then removed, and the number of insects in each com-partment was recorded. Each experiment was divided into four groups, wild-type P. absoluta adults with T. graminum, B. tabaci, N. tenuis, and T. pintoi, with 30 in-sects per group. All experiments were repeated four times and the behavioral response device was cleaned after each experiment. In Table 2, before the light-trapping experiment, wild LW2(-/-) strain, T. graminum, B. tabaci, N. tenuis, and T. pintoi were placed in a dark room for 1 h simultaneously, and the drawer board was opened. After entering the behavioral response area, 16 LED lights (325 ± 5 nm, 365 ± 5 nm, 375 ± 5 nm, 385 ± 5 nm, 390 ± 5 nm, 395 ± 5 nm, 400 ± 5 nm, 405 ± 5 nm, 415 nm ± 5, 420 ± 5 nm, 430 ± 5 nm, 450 ± 5 nm, 460 ± 5 nm, 490 ± 5 nm, 520 ± 5 nm, and 590 ± 5 nm) with different wavelengths were turned on at the same time, and the partition of the cubicle was opened so that the test insects could enter the cubicle according to their own preferences. The black velvet was then removed, and the number of insects in each compartment was recorded. Each experiment was divided into four groups, LW2(-/-) strain adults with T. graminum, B. tabaci, N. tenuis, and T. pintoi, with 30 insects per group. All experiments were repeated four times and the behavioral response device was cleaned after each experiment. In Table 3, before the light-trapping experiment, wild BL(-/-) strain, T. graminum, B. tabaci, N. tenuis, and T. pintoi were placed in a dark room for 1 h simultaneously, and the drawer board was opened. After entering the behavioral response area, 16 LED lights (325 ± 5 nm, 365 ± 5 nm, 375 ± 5 nm, 385 ± 5 nm, 390 ± 5 nm, 395 ± 5 nm, 400 ± 5 nm, 405 ± 5 nm, 415 nm ± 5, 420 ± 5 nm, 430 ± 5 nm, 450 ± 5 nm, 460 ± 5 nm, 490 ± 5 nm, 520 ± 5 nm, and 590 ± 5 nm) with different wavelengths were turned on at the same time, and the partition of the cubicle was opened so that the test insects could enter the cubicle according to their own preferences. The black velvet was then removed, and the number of insects in each compartment was recorded. Each experiment was divided into four groups, BL(-/-) strain adults with T. graminum, B. tabaci, N. tenuis, and T. pintoi, with 30 insects per group. All experiments were repeated four times and the behavioral response device was cleaned after each experiment.

Each set of experiments in Tables 1, 2, and 3 had four replicates and used the same wavelength.

We found that the results obtained by T. graminum, B. tabaci, N. tenuis, and T. pintoi were all similar, but there were differences among wild tomato leaf miner, LW2(-/-), and BL(-/-) strains, which may be related to the interaction between visual protein and insect visual system. The spectral specificity of LW2 gene is 500-600nm, so LW2 can regulate insects' preference for short wavelength. However, the wavelength range of BL gene is 400-500nm, and its mutation directly affects the wavelength trend behavior of insects.

Q7. Finally, I don’t feel a strong connection between the information given in the first figure and the rest of the paper, even though I totally agree that this information is important. But why do the authors draw the conclusion about the importance of the opsin protein(s) from the fact they are expressed in different parts of the body? It makes some sense but it not entirely clear. It’s also interesting to elaborate (maybe in the discussion) if the authors suggest that the opsins expressed extraocularly have a role in photoreception and attraction by light. Is there such evidence for the studied or related species?

Response: Thanks for your comments. Opsin proteins are essential for vision in insects. Light enters the insect's compound eye and stimulates retinal molecules embedded in rhodopsin, triggering isomerization of the retina and the downstream phototransduction cascade. Phthorimaea absoluta, as a nocturnal pest, can carry out a series of behavioral activities in low light environment at night, which we speculate is related to the expression of opsin. For example, in Apolygus lucorum, opsin genes (A1–3) were expressed highly not only in head but also in leg, wing, and mouthpart, indicating these opsins may execute some nonvisual functions,among these, the LW opsin (A1) showed the highest expression levels in the head tissue of adults, indicating that LW opsin may help the organism adapt to a low light environment (Gao et al., 2021). Therefore, we believe that the diversity of opsin expression may be related to its adaptation to light environment, and may not directly play a role in light perception and light attraction. Reference is as follows:

Gao, H.; Li, Y.; Wang, M.; Song, X.; Tang, J.; Feng, F.; Li, B. Identification and expression analysis of G protein-coupled receptors in the Miridae insect Apolygus lucorum. Front. Endocrinol. 2021, 12, 773669.

Q8. L21 “Long-wavelength” should be lowercase (long-wavelength)

Response: Thanks for your comments. We have changed " Long-wavelength " to " long-wavelength ", please check in line 22.

Q9. L65 “found that tomato leaf miner attracts black light” => “found that tomato leaf miner is attracted by black light”

Response: Thanks for your comments. We have changed " found that tomato leaf miner attracts black light " to " found that tomato leaf miner is attracted by black light ", please check in line 66.

Q10.L78-79: opsinS (as there may be more than one in each class)

Response: Thanks for your comments. We have changed " opsinS " to " opsins", please check in line78.

Q11.L88 “pest control In this study” : full stop missing

Response: Thanks for your comments. We have added a period, please check it on line 90.

Q12.L176 “Digoxin-labeled” should be “digoxin-labeled”

Response: Thanks for your comments. We have changed " Digoxin-labeled " to " digoxin-labeled ", please check in line 178.

Q13.Tables: please make species names italic

Response: Thanks for your comments. We have made the revision, please check it on lines 197, 210, 223, and 236.

Q14.L236 and below: “Opsin is highly expressed”: I would say opsins, as these are all different representatives of this protein group.

Response: Thanks for your comments. We have made the revision, please check it on lines 243, 246, 247, and 248.

Round 2

Reviewer 2 Report

Comments and Suggestions for Authors

First and foremost, I’d like to commend the authors for their work on the revision.

Second, while I’m satisfied with the responses to my concerns, I feel it’s a pity that some of them did not find its way to the actual manuscript, as I feel I’m not the only person who might have them.

So, could you please add the information about light sources, which was described in detail in the response, to the methods section of the manuscript?

In addition, it would be very beneficial to state clearly that the numbers in the tables are mean numbers of animals (not percent).

Author Response

Responses to reviewers 2

May 17, 2024

We have read and studied the reviewers’ comments and suggestions very carefully. We have revised our manuscript based on the reviewers' comments. The main corrections in the paper and the responses to the reviewer’s comments are as follows.

Responses to Reviewer 2

Comments to the Author:

This study explores the phototactic behavior of an important pest and green methods (i.e. not involving any pollutants) to trap them. In fact, the authors are too shy, as in fact they tested more species. The study is both interesting and important, and I enjoyed reading it. However, I feel it could benefit from getting more detailed in some places (see below).

Q1: First and foremost, I’d like to commend the authors for their work on the revision.

Response: Thanks for your comments.

Q2: Second, while I’m satisfied with the responses to my concerns, I feel it’s a pity that some of them did not find its way to the actual manuscript, as I feel I’m not the only person who might have them.

So, could you please add the information about light sources, which was described in detail in the response, to the methods section of the manuscript?

Response: Thanks for your comments. We have added the information about the light source described in detail in the reply to the method part of the manuscript as follows: The light source manufacturer comes from Shenzhen Fangpu Optoelectronics Co., Ltd., which provides light sources for the research and development of many university laboratories and enterprises. The lamp source is composed of integrated LED lamp beads of different powers, the chromatographic peak test method is FWHM, the lens power can be 60 degrees, 90 degrees, 120 degrees, and the voltage is 220, please check line from 145 to 150.

Q3: In addition, it would be very beneficial to state clearly that the numbers in the tables are mean numbers of animals (not percent).

Response: Thanks for your comments. We've added a table note below each table that reads as follows: the data in the table represent the average number of insects at different wavelengths, please check in lines 203, 217, 231, 245.

Thank you once again for your profound comments and questions. We hope our corrections meet your requirements. We are very grateful to you for your kind help in processing this manuscript. We look forward to seeing its publication in your journal soon.

With best regards,

Zhi-Chuang Lü

***********************************

Zhi-Chuang Lü

Department of Biological Invasions

Institute of Plant Protection

Chinese Academy of Agricultural Sciences

Beijing 100193

CHINA

Tel & Fax: +86 10 82109572

E-mail: lvzhichuang@caas.cn